# Comparison of Radiation Exposure between Endoscopic Ultrasound-Guided Hepaticogastrostomy and Hepaticogastrostomy with Antegrade Stenting

**DOI:** 10.3390/jcm11061705

**Published:** 2022-03-19

**Authors:** Mamoru Takenaka, Madan M. Rehani, Makoto Hosono, Tomohiro Yamazaki, Shunsuke Omoto, Kosuke Minaga, Ken Kamata, Kentaro Yamao, Shiro Hayashi, Tsutomu Nishida, Masatoshi Kudo

**Affiliations:** 1Department of Gastroenterology and Hepatology, Faculty of Medicine, Kindai University, Osakasayama 589-8511, Japan; chochiko.4kg@gmail.com (T.Y.); shunsuke.oomoto@gmail.com (S.O.); kousukeminaga@yahoo.co.jp (K.M.); ky11@leto.eonet.ne.jp (K.K.); yamaken_volvo@yahoo.co.jp (K.Y.); m-kudo@med.kindai.ac.jp (M.K.); 2Global Outreach for Radiation Protection Programme, Radiation Safety Committee, Massachusetts General Hospital, Boston, MA 02114, USA; madan.rehani@gmail.com; 3Department of Radiology, Faculty of Medicine, Kindai University, Osakasayama 589-8511, Japan; hosono@med.kindai.ac.jp; 4Department of Gastroenterology, Toyonaka Municipal Hospital, Toyonaka 560-8565, Japan; rexgistudy@gmail.com (S.H.); tnishida.gastro@gmail.com (T.N.); 5Department of Gastroenterology and Internal Medicine, Hayashi Clinic, Suita 565-0842, Japan

**Keywords:** fluoroscopy, ultrasound-guided hepaticogastrostomy, radiation exposure

## Abstract

Fluoroscopy forms an essential part of endoscopic ultrasound-guided hepaticogastrostomy (EUS-HGS) and hepaticogastrostomy with antegrade stenting (EUS-HGAS). To date, no study has assessed and compared radiation exposure between EUS-HGS and EUS-HGAS. This study aimed to compare the radiation exposure parameters between EUS-HGS and EUS-HGAS. This retrospective single-center cohort study included consecutive patients who underwent EUS-HGS or EUS-HGAS from October 2017 to March 2019. The air kerma (AK: mGy), kerma–area product (KAP: Gycm^2^), fluoroscopy time (FT: min), and procedure time (PT: min) were assessed and compared between the two procedures. Altogether, 45 and 24 patients underwent EUS-HGS and EUS-HGAS, respectively. The median AK, KAP, FT, and PT were higher in the EUS-HGAS group than in the EUS-HGS group. A comparison revealed no difference in the technical success rate, complications rate, adverse event occurrence rate, and re-intervention rate between both procedures. This is the first report in which radiation exposure was used as a comparative parameter between EUS-HGS and EUS-HGAS. This study revealed that radiation exposure is significantly higher in EUS-HGAS than in EUS-HGS. Increased awareness on radiation exposure is warranted among gastroenterologists so that they choose the procedure with lower radiation exposure in cases where both procedures are indicated.

## 1. Introduction

The usefulness of endoscopic ultrasound-guided biliary drainage (EUS-BD) has been reported for malignant biliary obstruction (MBO), in which transpapillary drainage by endoscopic retrograde cholangiopancreatography (ERCP) is impossible [1,2,3,4,5]. EUS-guided hepaticogastrostomy (EUS-HGS) is one of the EUS-BD techniques in which trans-gastrointestinal drainage is performed on the dilated intrahepatic bile duct due to MBO (Figure 1a). Additionally, the usefulness of EUS-guided antegrade stenting (EUS-AS) combined with EUS-HGS (EUS-HGAS) has also been reported (Figure 1b) [6,7,8].

In many cases of MBO, both EUS-HGS and EUS-HGAS may be indicated. Although many studies have compared these procedures, there is no consensus on which procedure is better [9,10,11]. In clinical practice, the selection of procedure is primarily based on the endoscopist’s discretion and hospital policy.

Moreover, both procedures are performed under fluoroscopy using EUS imaging for guidance. Fluoroscopy is essential for these procedures, and it is desirable to know the “radiation exposure” of a patient [12] and compare the dose parameters among different procedures.

The International Commission on Radiological Protection (ICRP) has provided the principle of “As Low As Reasonably Achievable”; thus, keeping radiation exposure low is important [13,14]. Furthermore, reducing patient exposure has a proportionate effect on staff exposure and thus the need to assess and optimize patient radiation exposure [15].

Although many studies have compared the usefulness of each EUS-BD procedure, focusing on the procedure’s technical and clinical success rates and the frequency of complications [9,10,11], no study has used radiation exposure as a comparative parameter. This fact suggests that gastroenterologists have low awareness of radiation exposure.

Hence, in this study, we compared the radiation exposure between EUS-HGAS and EUS-HGS.

## 2. Materials and Methods

The study protocol was approved by the Institutional Review Board of Kindai University (IRB No. R02-121). The study followed the provisions of the declaration of Helsinki, as revised in Fortaleza, Brazil, in 2013. All authors had full access to all the data in this study and accepted the responsibility for submitting it for publication.

### 2.1. Patients

This was a single-center, retrospective study conducted at the Faculty of Medicine, Kindai University from October 2017 to March 2019. During the study period, 45 EUS-HGS and 24 EUS-HGAS procedures were performed consecutively using the same fluoroscopy device with an over-couch X-ray tube (CUREVISTA17, Hitachi Co., Tokyo, Japan). A written informed consent including the understanding that data related to the procedure would be used for research was obtained from each patient before the procedure.

### 2.2. EUS-HGS and EUS-HGAS

All EUS-BD procedures were performed by experienced endoscopists who have completed a minimum of 300 endoscopic procedures annually for at least 5 years. The procedures were performed under conscious sedation using intravenous propofol and pethidine.

For EUS-HGS, an echoendoscope (GF-UCT260; Olympus Medical, Tokyo, Japan) was advanced into the stomach or jejunum to visualize the dilated left intrahepatic bile duct (IHBD) as a drainage target. After identifying an appropriate puncture route without interposing vessels, the IHBD was punctured under EUS guidance using a 19-gauge EUS-guided fine-needle aspiration (FNA) needle (SonoTip; Medi-Globe GmbH; Rohdorf, Germany, or EzShot3; Olympus Medical, Tokyo, Japan). The bile juice was aspirated, and the contrast medium was injected for cholangiography under fluoroscopy to evaluate the obstruction site. A 0.025-inch guidewire (VisiGlide2; Olympus Medical, Tokyo, Japan) of appropriate length was inserted into the IHBD, and the fistula was dilated to 4 mm using a balloon catheter (REN biliary dilation catheter; KANEKA, Osaka, Japan). Finally, a stent was deployed between the IHBD and the stomach or jejunum. A partially covered metal stent (Niti-S S-type Stent; Taewoong Medical, Seoul, Korea) or a EUS-BD dedicated plastic stent (TYPE-IT Stent; Gadelius Medical, Tokyo, Japan) [16] was used for transmural stenting (Figure 2). The thickness of the stent was selected from 8 mm or 10 mm based on the diameter of the bile duct in which the stent was placed, and the length of the stent was also selected from 8 cm or 10 cm depending on the site where the stent was placed.

EUS-HGAS was performed following a similar process as EUS-HGS until a guidewire was inserted into the IHBD. In EUS-HGS, the guidewire was inserted into the IHBD, but in EUS-HGAS, it was manipulated from IHBD to the duodenal lumen beyond the MBO site. A catheter replaced a EUS-FNA needle to manipulate the guidewire. After the fistula dilation to 4 mm using a balloon catheter (REN biliary dilation catheter; KANEKA, Osaka, Japan), antegrade stenting was performed under fluoroscopic guidance using an uncovered metal stent (BileRush Selective, Piolax, Yokohama, Japan). Then transmural stenting was performed under fluoroscopy (Figure 3).

The fluoroscopic system was well maintained, and its performance was assessed periodically by a qualified engineer. In both procedures, fluoroscopy was used to produce live imaging and essential static imaging.

### 2.3. Outcome Definitions

The primary aim of this study was to compare the radiation exposure parameters between EUS-HGS and EUS-HGAS.

The evaluation parameters used for radiation exposure were the standard factors available in most fluoroscopy machines and recommended by the International Commission on Radiological Protection [13,17]. The radiation exposure evaluation items used in this study were air kerma at the patient entrance reference point (AK: mGy), kerma–area product (KAP: Gycm^2^), and fluoroscopy time (FT: min). Each procedural detail, including procedure time (PT: min), was recorded in a database and updated after every study. The values of AK, KAP, FT, and PT for EUS-HGS and EUS-HGAS were assessed and compared.

The second aim was to compare the procedures for technical success rate, clinical success rate, adverse event occurrence rate, and re-intervention rate.

Technical success was defined as successful stent deployment between the IHBD and the stomach or jejunum in EUS-HGS cases, and both successful antegrade stent deployment beyond the MBO site and stent deployment between the IHBD and the stomach or jejunum in EUS-HGAS cases.

Clinical success was defined as a decrease in the bilirubin concentration to <40% of the pretreatment value within 2 weeks. The rate of adverse events, such as peritonitis, bleeding, stent migration, and hyperamylasaemia, was also assessed. Re-intervention was defined as any endoscopic, surgical, or percutaneous procedure required to improve symptoms after the stent placement.

### 2.4. Statistical Analysis

All continuous variables were expressed as the median with interquartile range (IQR). Categorical variables were expressed as numbers in each category or frequency, whereas continuous variables were compared using Student’s *t*-test or Dunn’s multiple comparison test. Categorical variables were compared using the chi-square test or Fisher’s exact test when appropriate. A two-tailed *p*-value < 0.05 was considered statistically significant. All statistical analyses were performed using JMP software (ver. 12; SAS Institute, Inc., Cary, NC, USA).

## 3. Results

### 3.1. Baseline Characteristics

The baseline characteristics of all patients who underwent EUS-HGS and EUS-HGAS during this study period are shown in Table 1.

The median age (IQR) was 73 (65–77) years in the EUS-HGS group (33 men and 12 women) and 71.5 (68–84) years in the EUS-HGAS group (18 men and 6 women). The disease occurrence for each procedure type (EUS-HGS vs. ERCP-HGAS) was as follows: pancreatic cancer, 15 (33.3%) vs. 12 (50.0%); gastric cancer (lymph node metastasis), 10 (22.2%) vs. 8 (33.3%); cholangiocarcinoma, 6 (13.3%) vs. 0 (0.0%), and hepatocellular carcinoma, 6 (13.3%) vs. 0 (0.0%); and others, 8 (17.8%) vs. 4 (16.7%) (*p* = 0.08).

The reasons for EUS-BD (EUS-HGS vs. ERCP-HGAS) were as follows: failed biliary cannulation, 21 (46.7%) vs. 10 (41.7%); surgical anatomy, 18 (40.0%) vs. 8 (33.3%), and duodenal obstruction, 6 (13.3%) vs. 6 (25.0%) (*p* = 0.47).

### 3.2. Radiation Exposure

The radiation dose (AK/KAP) and time factors (FT/PT) in the EUS-HGS and EUS-HGAS groups are shown in Table 2.

In terms of the radiation dose, AK was significantly higher (*p* = 0.0014) in the EUS-HGAS group than in the EUS-HGS group (135.0 vs. 88.4 mGy [median]). The KAP was significantly higher (*p* = 0.0006) in the EUS-HGAS group than in the EUS-HGS group (33.3 vs. 23.0 Gycm^2^ [median]).

Moreover, in terms of time, FT was significantly longer (*p* < 0.0001) in the EUS-HGAS than in the EUS-HGS group (33.7 vs. 15.8 min [median]). The PT was significantly longer (*p* < 0.0001) in the EUS-HGAS group than in the EUS-HGS group (51.0 vs. 29.0 min [median]) (Figure 4).

### 3.3. The Outcomes of Each Procedure

The outcomes of each procedure are shown in Table 3. There was no difference in the technical success rate between the two procedures (95.6% in the EUS-HGS group vs. 83.3% in the EUS-HGAS group; *p* = 0.17). Upon examining only successful cases, there was no significant difference in the clinical success rate between the two procedures (93.0% in the EUS-HGS group vs. 90.0% in the EUS-HGAS group; *p* = 0.65). Although there was no significant difference, the EUS-HGAS group tended to have a higher frequency of complications than the EUS-HGS group (*p* = 0.06). Peritonitis occurred in 4.7% of patients in the EUS-HGS group and 10.0% in the EUS-HGAS group. Bleeding occurred only in the EUS-HGS group (4.7%), and hyperamylasaemia occurred only in the EUS-HGAS group (20.0%). Re-intervention was performed in 20.9% and 10.0% of the patients in the EUS-HGS and EUS-HGAS groups, respectively (*p* = 0.48).

## 4. Discussion

This is the first report comparing EUS-HGS and EUS-HGAS in terms of radiation exposure. It is also the first report comparing radiation exposure among EUS-BD-related procedures. This comparative study revealed that radiation exposure is significantly higher in EUS-HGAS than in EUS-HGS. Since cumulative radiation exposures cause various health hazards, it is recommended to choose lower radiation exposure in cases where both EUS-HGS and EUS-HGAS are equally indicated on clinical grounds.

For biliary drainage, EUS-HGS has an advantage over EUS-HGAS as it requires fewer procedural steps and, therefore, less time to perform. However, when stent occlusion occurs, re-intervention using the transgastrointestinal drainage route is sometimes challenging, although various techniques have been reported [18,19,20,21,22]. In EUS-HGS, the re-intervention difficulty is a clinical problem.

Contrastingly, EUS-HGAS has two drainage routes, transgastrointestinal and transpapillary; thus, even if one is obstructed, the other ensures less biliary atresia. The rate of re-intervention was reported to be lower in EUS-HGAS [9]. This is one of the reasons why endoscopists place not only antegrade transpapillary stents, but also transgastrointestinal stents. The results of this study were similar to our findings. Additionally, even if a re-intervention is necessary, the transpapillary drainage route is relatively easier to approach. However, EUS-HGAS involves more procedural steps than EUS-HGS and needs two stents, resulting in longer procedural durations and higher costs.

Thus, both procedures have advantages and disadvantages, and there is no clear consensus regarding superiority. No clear difference in clinical success is reported between these procedures, similar to this study’s results [9]. In cases where both EUS-HGAS and EUS-HGS can be accepted, the selected procedure is decided by the operator and facility policy. However, the current findings showing a significantly high radiation exposure in EUS-HGAS than in EUS-HGS, because of longer procedure and fluoroscopy durations, may impact procedure choice.

The radiation doses of EUS-HGAS and EUS-HGS revealed in this study were not high enough to cause a significant risk of carcinogenesis or skin damage in a single session. As EUS-BD is not usually performed more than once on the same patient, radiation exposure is unlikely to be seen as a major problem. However, endoscopists should be aware that patients are undergoing an increasing number of computed tomography (CT) examinations and other fluoroscopically guided procedures, resulting in alarmingly high cumulative doses that have never been witnessed before [23,24]. It is rather rare to find a patient with chronic disease who may not be undergoing ionizing radiation procedures. Therefore, there is a need for higher awareness among gastroenterologists on radiation risks.

As indicated above, the higher the patient radiation dose during an examination, the higher the radiation exposure to the medical personnel. If the facility’s radiation protection measures are inadequate and the medical personnel is unaware of the radiation exposure, then cumulative radiation exposure will quickly reach levels that can cause health effects.

This lack of awareness of radiation exposure among gastroenterologists is an important issue that needs to be addressed to improve this situation. The American Society for Gastrointestinal Endoscopy recommends applying the frequency with which fluoroscopy time and radiation exposure are measured and documented as a quality indicator for ERCP [25]. The European Society of Gastrointestinal Endoscopy guidelines on radiation protection in digestive endoscopy recommend establishing a standard radiation dose for ERCP [26]. However, there are only a few reports on radiation exposure in ERCP, even fewer on EUS-BD [27,28,29], and, of course, none on EUS-HGAS and EUS-HGS. Therefore, the results of this study will have a significant impact on a gastroenterologist.

EUS-BD has undergone various developments over the past 20 years since it was first reported in 2001 [30], but not much has been done with the concept to reduce radiation dose. Currently, EUS-BD is mainly carried out using non-dedicated devices, such as EUS-FNA needles, ERCP catheters, and guidewires. It is hoped that all devices used for EUS-BD will be improved to reduce radiation exposure in each case.

The present study has several limitations. First, it was a single-center retrospective analysis with a small number of patients. Second, there is a bias in the procedure time. Our institution is experienced in EUS-BD, but longer examination and fluoroscopy times during EUS-BD are expected in less experienced institutions. Thirdly, the radiation exposure measurements in this study were taken with a single fluoroscopy device. There are many different types of fluoroscopy equipment and each hospital uses different fluoroscopy equipment. Therefore, it is not clear whether the trend of the present results will be confirmed in all institutions. Further analysis of radiation dose measurements using different types of fluoroscopy equipment at multiple sites will be helpful. However, even with these limitations, the results of this study will have an impact on many gastroenterologists who perform EUS-BD. It will be meaningful if this study increases awareness among gastroenterologists who have not paid much attention to radiation exposure.

In conclusion, overall, this analysis revealed that the radiation exposure was significantly higher during EUS-HGAS than during EUS-HGS. Considering the cumulative effect of radiation exposure and the fact that patients undergo many other examinations, such as CT, it is desirable to make procedure choices with lower radiation exposure, keeping in mind the impact on not only the endoscopists but also the assistants and nurses who are exposed to radiation as well.

## Figures and Tables

**Figure 1 jcm-11-01705-f001:**
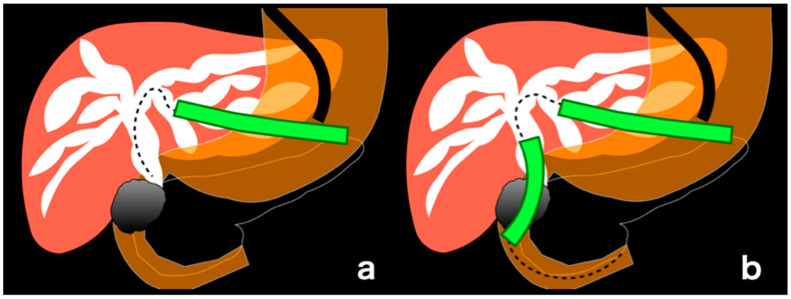
(**a**). Schematic drawing of endoscopic ultrasound-guided hepaticogastrostomy (EUS-HGS); In EUS-HGS, one stent (green) is placed transgastrointestinal. (**b**). Schematic drawing of EUS-guided antegrade stenting (EUS-AS) combined with EUS-HGS (EUS-HGAS); In EUS-HGAS, two stents (green), transgastrointestinal and transpapillary, are placed.

**Figure 2 jcm-11-01705-f002:**
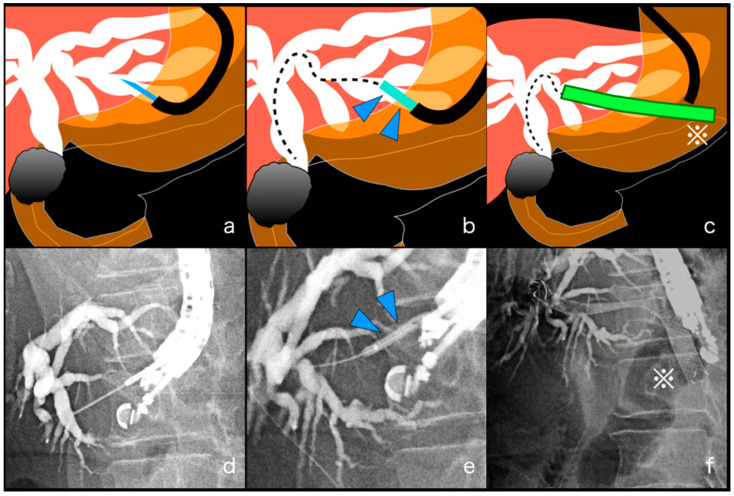
Procedure process of endoscopic ultrasound-guided hepaticogastrostomy (EUS-HGS). (**a**,**d**). The dilated biliary duct was punctured under EUS guidance using a 19-gauge EUS-guided fine-needle aspiration needle and the contrast medium was injected for cholangiography under fluoroscopy. (**b**,**e**). The fistula was dilated to 4 mm using a balloon catheter (blue, arrowhead) under fluoroscopy. (**c**,**f**). Finally, a stent (green, ※) was deployed between the dilated bile duct and the stomach or jejunum.

**Figure 3 jcm-11-01705-f003:**
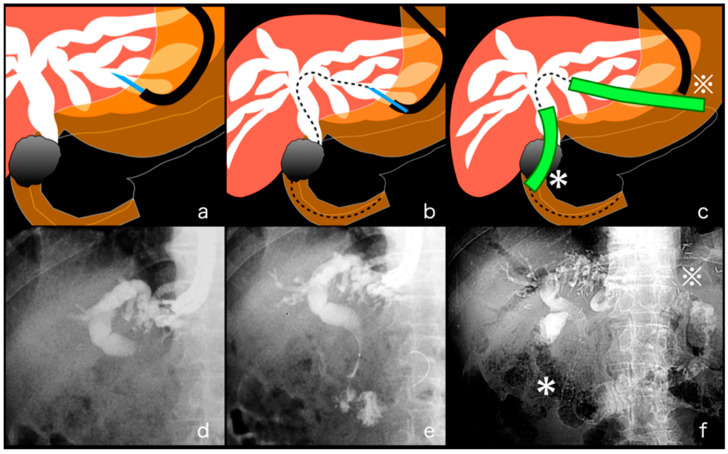
Procedure process of EUS-guided antegrade stenting (EUS-AS) combined with hepaticogastrostomy (EUS-HGAS). (**a**,**d**) The dilated biliary duct was punctured under EUS guidance using a 19-gauge EUS-guided fine-needle aspiration needle and the contrast medium was injected for cholangiography under fluoroscopy. (**b**,**e**) The guidewire was manipulated from the dilated bile duct to the duodenal lumen beyond the malignant biliary obstruction (MBO) site. (**c**,**f**) Both antegrade stenting (green, *) through MBO and a stent (green, ※) between the dilated bile duct and the stomach or jejunum were deployed.

**Figure 4 jcm-11-01705-f004:**
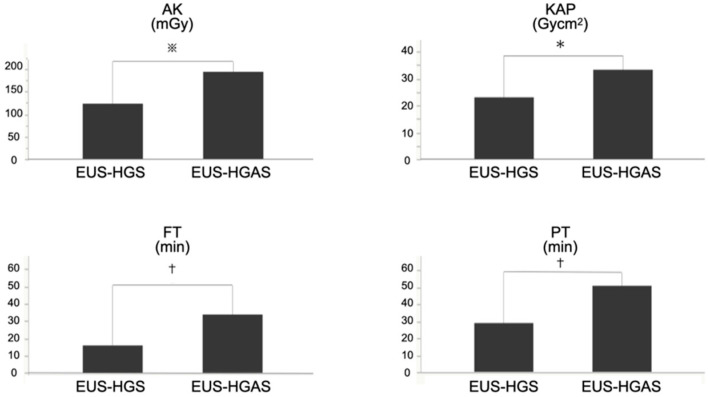
The radiation dose (AK/KAP) and time factors (FT/PT) in the EUS-HGS and EUS-HGAS groups. The median AK, KAP, FT, and PT were higher in the EUS-HGAS group than in the EUS-HGS group. (※ *p* = 0.0014, *
*p* = 0.0006, † *p* < 0.0001). AK: air kerma, KAP: kerma–area product, FT: fluoroscopy time, PT: procedure time, EUS-HGS: endoscopic ultrasound-guided hepaticogastrostomy, EUS-HGAS: EUS-guided antegrade stenting (EUS-AS) combined with hepaticogastrostomy.

**Table 1 jcm-11-01705-t001:** Patient characteristics.

	EUS-HGS (*n* = 45)	EUS-HGAS (*n* = 24)	*p*-Value
Median age (IQR)	73 (65–77)	71.5 (68–84)	0.34
Male/female	33/12	18/6	1.00
Disease	*n* (%)	*n* (%)	0.08
Pancreatic cancer	15 (33.3)	12 (50.0)	
Gastric cancer (lymph node metastasis)	10 (22.2)	8 (33.3)	
Cholangiocarcinoma	6 (13.3)	0 (0.0)	
Hepatocellular carcinoma	6 (13.3)	0 (0.0)	
Others	8 (17.8)	4 (16.7)	
Reason for EUS-BD	*n* (%)	*n* (%)	0.47
Failed biliary cannulation	21 (46.7)	10 (41.7)	
Surgical anatomy	18 (40.0)	8 (33.3)	
Duodenal obstruction	6 (13.3)	6 (25.0)	

*p* < 0.05 was considered statistically significant. EUS-HGS: endoscopic ultrasound-guided hepaticogastrostomy, EUS-HGAS: endoscopic ultrasound-guided antegrade stenting combined with endoscopic ultrasound-guided hepaticogastrostomy, IQR: interquartile range, EUS-BD: endoscopic ultrasound-guided drainage.

**Table 2 jcm-11-01705-t002:** Comparison of the radiation dose parameter (AK/DAP) and time factors (FT/PT) between EUS-HGS and EUS-HGAS.

	EUS-HGS (*n* = 45)	EUS-HGAS (*n* = 24)	*p*-Value
**Radiation dose**			
AK (mGy), median (IQR)	123.4 (77.4–180.5)	194.4 (138.8–264.7)	0.0014
KAP (Gycm^2^), median (IQR)	23.0 (17.5–30.7)	33.3 (28.0–40.3)	0.0006
**Time**			
FT (min), median (IQR)	15.8 (11.7–19.7)	33.7 (24.8–37.6)	<0.0001
PT (min), median (IQR)	29.0 (24.5–36)	51.0 (48–69)	<0.0001

*p* < 0.05 was considered statistically significant. EUS-HGS: endoscopic ultrasound-guided hepaticogastrostomy, EUS-HGAS: endoscopic ultrasound-guided antegrade stenting combined with endoscopic ultrasound-guided hepaticogastrostomy, AK: air kerma, DAP: dose–area product, FT: fluoroscopy time, PT: procedure time, IQR: interquartile range.

**Table 3 jcm-11-01705-t003:** The success and adverse event occurrence rates of EUS-HGS and EUS-HGAS.

	EUS-HGS (*n* = 45)	EUS-HGAS (*n* = 24)	*p*-Value
Technical success rate (%)	95.6 (43/45)	83.3 (20/24)	0.17
Clinical success rate ^※^ (%)	93.0 (40/43)	90.0 (18/20)	0.65
Adverse event occurrence rate ^※^ (%)	9.3 (4/43)	30.0 (6/20)	0.06
Peritonitis	4.7 (2/43)	10.0 (2/20)	
Bleeding	4.7 (2/43)	0.0 (0/20)	
Stent migration	0.0 (0/43)	0.0 (0/20)	
Hyperamylasemia	0.0 (0/43)	20.0 (4/20)	
Re-intervention occurrence rate ^※^ (%)	20.9 (9/43)	10.0 (2/20)	0.48

*p* < 0.05 was considered statistically significant. ^※^ Examining only successful cases. EUS-HGS: endoscopic ultrasound-guided hepaticogastrostomy, EUS-HGAS: endoscopic ultrasound-guided antegrade stenting combined with endoscopic ultrasound-guided hepaticogastrostomy.

## Data Availability

All the data used for this analysis can be confirmed at any time.

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
