# Peer review of "Comparison of Radiation Exposure between Endoscopic Ultrasound-Guided Hepaticogastrostomy and Hepaticogastrostomy with Antegrade Stenting"

_jcm, 2022, doi:10.3390/jcm11061705_

Round 1
Reviewer 1 Report
- Introduction: write malignant biliary obstruction before the first time that appears its abbreviation MBO
- In figure 1, instead of “schema” it could be more appropriate: schematic drawing of endoscopic ultrasound-guided hepaticogastrostomy (EUS-HGS); b. schematic drawing of EUS-guided antegrade stenting (EUS-AS) combined with EUS-HGS (EUS-HGAS).
- Material and Methods: “fistula dilation was then carried out using a balloon catheter”. It would be very useful to know how many millimeters the balloon was inflated.
In the same way “finally, a stent was deployed between the IHBD and the stomach or jejunum” it should be useful for endoscopists reading the paper to know, if possible, length and diameter of such stents
In the description of antegrade stenting the tract was dilated after passing the antegrade stent for insertion of the hepatic-gastric stent:
“Subsequently, the puncture tract was dilated using a balloon catheter, and transmural stenting was performed under fluoroscopy (Figure 3).”(105-107)
I wonder if the antegrade stent passed easily through the liver without previous dilation.
- Results: In patients with gastric cancers having also jaundice, I suppose it was because of lymph nodes compression. Please, indicate this fact.
“The peritonitis occurred in 4.7% of patients 197”. Delete “The” and write only “peritonitis occurred…”
- Discussion: it should be very helpful for endoscopists to comment why in the antegrade methods it is not enough with the transduodenal stent, and a second stent, as in the hepatico-gastrostomy method had to be inserted.
Author Response
Reviewer 1
- Introduction: write malignant biliary obstruction before the first time that appears its abbreviation MBO
【Response】
Thanks for pointing that out. It can be found in the second line of the Introduction, please confirm.
- In figure 1, instead of “schema” it could be more appropriate: schematic drawing of endoscopic ultrasound-guided hepaticogastrostomy (EUS-HGS); b. schematic drawing of EUS-guided antegrade stenting (EUS-AS) combined with EUS-HGS (EUS-HGAS).
【Response】
Thanks for pointing that out. I have revised the text as such, please confirm.
- Material and Methods: “fistula dilation was then carried out using a balloon catheter”. It would be very useful to know how many millimeters the balloon was inflated.
In the same way “finally, a stent was deployed between the IHBD and the stomach or jejunum” it should be useful for endoscopists reading the paper to know, if possible, length and diameter of such stents
In the description of antegrade stenting the tract was dilated after passing the antegrade stent for insertion of the hepatic-gastric stent:
“Subsequently, the puncture tract was dilated using a balloon catheter, and transmural stenting was performed under fluoroscopy (Figure 3).”(105-107)
I wonder if the antegrade stent passed easily through the liver without previous dilation.
【Response】
I apologize for the confusion caused by the insufficient description. I have followed your suggestion and revise the text as follows.
・and the fistula was dilated to 4 mm using a balloon catheter
・The thickness of the stent was selected from 8 mm or 10 mm based on the diameter of the bile duct in which the stent was placed, and the length of the stent was also selected from 8 cm or 10 cm depending on the site where the stent was placed.
・After the fistula dilation to 4 mm using a balloon catheter (REN biliary dilation catheter; KANEKA, Osaka, Japan), antegrade stenting was performed under fluoroscopic guidance using an uncovered metal stent (BileRush Selective, Piolax, Yokohama, Japan). Then transmural stenting was performed under fluoroscopy (Figure 3).
- Results: In patients with gastric cancers having also jaundice, I suppose it was because of lymph nodes compression. Please, indicate this fact.
“The peritonitis occurred in 4.7% of patients 197”. Delete “The” and write only “peritonitis occurred…”
【Response】
As you pointed out, the cause of jaundice in gastric cancer cases was compression due to lymph node metastasis, so I described it as such.
- Discussion: it should be very helpful for endoscopists to comment why in the antegrade methods it is not enough with the transduodenal stent, and a second stent, as in the hepatico-gastrostomy method had to be inserted.
【Response】
Thank you for important point out.
The advantage of having two stents is stated in the text as follows.
“Contrastingly, EUS-HGAS has two drainage routes, trans-gastrointestinal and transpapillary; thus, even if one is obstructed, the other ensures less biliary atresia. The rate of re-intervention was reported to be lower in EUS-HGAS (9).”
Therefore, after this sentence, I stated that “This is one of the reasons why endoscopists place not only antegrade transpapillary stents, but also trans-gastrointestinal stents.”
Reviewer 2 Report
The authors present a study wherein they compared the radiation exposure differences between patients undergoing EU-HGS or EU-HGAS and found that the procedure with Antegrade stenting had significantly higher exposure to radiation based on parameters such as air kerma, dose area product/KAP etc.
This is useful in determining whether EU-HGS should/can be the preferred over EU-HGAS procedure (based on lower radiation exposure).
The authors also point out to the limitations of the study including small cohort of patients, single centre data collection that may or may not be comparable to data from other institutions based on different fluoroscopy times.
Authors should compare the data from another institution of an equal sized cohort and see if similar trend was followed for radiation exposure as reported in the present study.
The study is impactful and should be expanded to data from more patients and institutions.
Author Response
Reviewer 2
The authors present a study wherein they compared the radiation exposure differences between patients undergoing EU-HGS or EU-HGAS and found that the procedure with Antegrade stenting had significantly higher exposure to radiation based on parameters such as air kerma, dose area product/KAP etc.
This is useful in determining whether EU-HGS should/can be the preferred over EU-HGAS procedure (based on lower radiation exposure).
The authors also point out to the limitations of the study including small cohort of patients, single centre data collection that may or may not be comparable to data from other institutions based on different fluoroscopy times.
Authors should compare the data from another institution of an equal sized cohort and see if similar trend was followed for radiation exposure as reported in the present study.
The study is impactful and should be expanded to data from more patients and institutions.
【Response】
As you pointed out, it needs to be verified whether the trend identified in this study is similar in other facilities. However, since this is the first comparison, we cannot compare it with previous reports.
We are currently conducting a prospective multi-center study, so we will verify the results there and report back.
Thank you very much for your suggestions and advice.